# The Integration of Genome-Wide DNA Methylation and Transcriptomics Identifies the Potential Genes That Regulate the Development of Skeletal Muscles in Ducks

**DOI:** 10.3390/ijms242015476

**Published:** 2023-10-23

**Authors:** Yinglin Lu, Jing Zhou, Fan Li, Heng Cao, Xingyu Zhang, Debing Yu, Zongliang He, Hongjie Ji, Kunpeng Lv, Guansuo Wu, Minli Yu

**Affiliations:** 1Department of Animal Genetics, Breeding and Reproduction, College of Animal Science and Technology, Nanjing Agricultural University, Nanjing 210095, China; luyinglin@njau.edu.cn (Y.L.); 2022805141@stu.njau.edu.cn (J.Z.); 2022805140@stu.njau.edu.cn (F.L.); 2021105032@stu.njau.edu.cn (H.C.); 2023105039@stu.njau.edu.cn (X.Z.); yudebing@njau.edu.cn (D.Y.); 2Nanjing Institute of Animal Husbandry and Poultry Science, Nanjing 210036, China; hezongliang542@126.com (Z.H.); 18251837056@163.com (H.J.); 18852006645@163.com (K.L.); njahps@163.com (G.W.)

**Keywords:** DNA methylation, embryonic ducks, RNA-seq, skeletal muscle development, WGBS

## Abstract

DNA methylation is a pivotal epigenetic regulatory mechanism in the development of skeletal muscles. Nonetheless, the regulators responsible for DNA methylation in the development of embryonic duck skeletal muscles remain unknown. In the present study, whole genome bisulfite sequencing (WGBS) and transcriptome sequencing were conducted on the skeletal muscles of embryonic day 21 (E21) and day 28 (E28) ducks. The DNA methylation pattern was found to fall mainly within the cytosine-guanine (CG) context, with high methylation levels in the intron, exon, and promoter regions. Overall, 7902 differentially methylated regions (DMRs) were identified, which corresponded to 3174 differentially methylated genes (DMGs). By using integrative analysis of both WGBS with transcriptomics, we identified 1072 genes that are DMGs that are negatively associated with differentially expressed genes (DEGs). The gene ontology (GO) analysis revealed significant enrichment in phosphorylation, kinase activity, phosphotransferase activity, alcohol-based receptors, and binding to cytoskeletal proteins. The Kyoto Encyclopedia of Genes and Genomes (KEGGs) analysis showed significant enrichment in MAPK signaling, Wnt signaling, apelin signaling, insulin signaling, and FoxO signaling. The screening of enriched genes showed that hyper-methylation inhibited the expression of *Idh3a*, *Got1*, *Bcl2*, *Mylk2*, *Klf2*, *Erbin*, and *Klhl38*, and hypo-methylation stimulated the expression of *Col22a1*, *Dnmt3b*, *Fn1*, *E2f1*, *Rprm*, and *Wfikkn1*. Further predictions showed that the CpG islands in the promoters of *Klhl38*, *Klf2*, *Erbin*, *Mylk2*, and *Got1* may play a crucial role in regulating the development of skeletal muscles. This study provides new insights into the epigenetic regulation of the development of duck skeletal muscles.

## 1. Introduction

Ensuring the supply of high-quality animal protein requires the highly efficient development of husbandry in the poultry industry, which includes chickens, ducks, and geese [1,2]. The production and quality of muscle during growth periods are influenced by the embryonic stages [3]. Pectoral muscles (PMs) in ducks showed arrested or even reduced growth during late embryonic development [4]. Myoblasts lose their ability to divide and initiate myogenic differentiation at the late embryonic stage [4,5]. However, the molecular mechanisms that underlie the development of skeletal muscles in ducks during the embryonic stages have not been studied systematically.

The development of skeletal muscles is regulated not only at the genomic level but also by the epigenetic regulation of DNA methylation and demethylation [6,7]. DNA methylation has important effects on meat quality between Chinese indigenous pig breeds and Western commercial pig breeds, which provides a unique model for analyzing the regulation of DNA methylation in meat quality [8]. Muscle regulatory factors (MRFs) are responsible for myogenesis, including the directional differentiation of myoblast cells into myotubes, the formation of myofibers, and the regulation of the expression of muscle-specific genes [9,10,11]. The growth, maintenance, and regeneration of skeletal muscles involve widespread epigenetic reprogramming, and the DNA methylation-mediated regulation of MRFs is required for myoblast cell differentiation [5]. Genome-wide DNA methylation plays an important role in regulating functional gene expression and muscle growth at key developmental stages in sheep [12,13]. During the final stages of embryonic development, the fusion of myoblasts into muscle fibers occurs [12]. A recent study revealed that DNA methylation plays a crucial role in porcine prenatal myogenesis and the postnatal growth of skeletal muscles [14]. An early study revealed that the DNA demethylation of myogenic genes contributed to the developmental differences in leg muscle (LM) between Wuzong and Shitou geese at the embryonic stage [9].

The proliferation and apoptosis of satellite cells during the development of skeletal muscles in chicken embryos were regulated by the DNA methylation of *Cfl2* according to whole genome bisulfite sequencing (WGBS) analysis [1]. In addition, the DNA methylation/*Sp*1/*Igf2bp3* axis was involved in porcine skeletal myogenesis [15]. By using a combination of WGBS and RNA-Seq technologies, it was found that *Cacna2d2* inhibited the JNK/MAPK signaling pathway through DNA methylation and, thus, inhibited porcine myogenic differentiation [16]. However, there have been few attempts to research the dynamics of genome-wide DNA methylation and the connection between the DNA methylome and the transcriptome during the development of skeletal muscles in duck embryos.

High genome-wide 5hmC levels in skeletal muscles at embryonic day 21 (E21) were observed, and it decreased sharply at embryonic day 28 (E28) in our previous study [17]. In order to further investigate the regulation of DNA methylation in duck skeletal muscles, WGBS and RNA-seq were conducted to reveal the profiles of the DNA methylome and transcriptomes. In the present study, the integrated association analysis of DNA methylation and gene expression was performed to screen for the critical genes and signaling pathways related to DNA methylation. This study will provide valuable information for exploring DNA methylation and the potential regulatory mechanisms involved in the development of duck skeletal muscles.

## 2. Results

### 2.1. DNA Methylation Pattern in Duck Skeletal Muscle

The skeletal muscles of duck embryos (E21 and E28) were collected for WGBS sequencing. WGBS was correlated with RNA-seq to screen for the essential genes and to validate their expression through RT-qPCR (Figure 1A). An examination of Bismark’s methylation sites revealed that the percentage of DNA methylation in the cytosine-guanine (CG) context ranged from 88.54% to 89.44% (Figure 1B). In comparison, DNA methylation in the CHG context fluctuated between 2.61% and 2.78%, and in the CHH context, it ranged from 7.95% to 8.68% (Figure 1B). The C loci of different contexts were analyzed for methylation density and genome-wide methylation levels. The genome methylation density was significantly different in the non-CG context, but it was not significantly different in the CG context. The genome methylation density in the CG context was >75% (Figure 1C). In contrast, the genome-wide methylation levels were not significantly different in the CG context and non-CG context, and the genome-wide methylation level in the CG context was >0.50 (Figure 1D).

The level of DNA methylation of duck skeletal muscles in various genomic functional elements revealed that there were different methylation levels in the three contexts. The DNA methylation level in the CG context was highest in the intron, exon, and 3′-untranslated region (3′UTR), followed by the promoter region, and the 5′-untranslated region (5′UTR) had the lowest levels (Figure 1E). In the CHG and CHH contexts, the promoter regions had higher DNA methylation levels, and the rest of the functional elements had lower DNA methylation levels (Figure 1E). The DNA methylation level was highest in the gene body, followed by Upstream2K and Downstream2K (Figure 1F). The distal of UPstream2K had the highest level of DNA methylation, and the proximal near the gene body had the lowest methylation level (Figure 1F).

### 2.2. Identification of Differentially Methylated Regions (DMRs)

A total of 3421 differentially methylated genes (DMGs) were found by comparing the detected DMRs with the duck reference genome (GCA_015476345.1, NCBI) (Figure 2A). Among them, 3174 genes were differentially methylated in the CG context, 224 genes were found in the CHH context, and 207 genes were found in the CHG context (Figure 2A). Their lengths were distributed primarily between 100–200 bp in the CG context (Figure 2B). The distribution of methylation levels in the DMRs indicated that the average methylation level of duck skeletal muscle was between 0.50 and 0.85 in E21 and E28, with no significant difference (Figure 2C). Without considering other regions, the DMR analysis revealed that the intron region had the most DMRs, followed by the exon and promoter regions (Figure 2D).

### 2.3. Functional Enrichment of Genes with Methylation

In order to enhance the comprehension of the potential functions of genes and pathways enriched in DMRs, the top 30 gene ontology (GO) terms were enriched significantly in the CG context. It showed that the DMGs were primarily enriched in biological processes, which included the regulation of metabolic processes, RNA biosynthesis, gene expression regulation, and cellular adhesion (Figure 3). Concurrently, the GO terms of molecular function were primarily enriched in those activities that regulated transcription and DNA binding (Figure 3A). The Kyoto Encyclopedia of Genes and Genomes (KEGGs) enrichment pathway analysis indicated that the DMGs were mostly enriched in metabolic signaling, MAPK signaling, adhesion, the regulation of actin cytoskeleton, Wnt signaling, autophagy, and mTOR signaling (Figure 3B).

### 2.4. Association Analysis of DNA Methylation and Gene Expression

The relevant analysis of DMGs and differentially expressed genes (DEGs) indicated that 2071 overlapping genes were enriched in the CG context, 81 in the CHH context, and 77 in the CHG context (Figure 4A). The DMGs in the CG context were categorized into hyper-methylated and hypo-methylated genes. Additionally, the DEGs were classified into highly and lowly expressed genes. The portion in which the DMGs were negatively correlated with the DEGs represents the most significant proportion. Among them, there were 642 genes with a down-regulation of the methylation level and an up-regulation of gene expression. In comparison, 681 genes were up-regulated in terms of the methylation level and down-regulation in terms of gene expression (Figure 4B).

### 2.5. The Key Pathways Enriched the DMGs Negatively Associated with DEGs

The GO enrichment analysis in the CG context indicated the enrichment of phosphorylation, protein phosphorylation, kinase activity, phosphotransferase activity, alcohol group receptor, transferase activity, phosphorus-containing group transfer, and cytoskeletal protein binding (Figure 5A). In turn, the KEGGs pathway enrichment analysis revealed that these genes were enriched significantly in MAPK signaling, Wnt signaling, apelin signaling, insulin signaling, and FoxO signaling (Figure 5B). The screening of the enriched genes revealed that *Bcl2*, *Klhl38*, *Klf2*, *Erbin*, *Dnmt3b*, *Mylk2*, *Got1*, *Idh3a, Fn1*, *Col22a1*, *E2f1*, *Rprm,* and *Wfikkn1* were all associated closely with the formation and hypertrophy of myofibers in ducks (Table 1).

### 2.6. Identification of Genes Implicated in the Development of Skeletal Muscles

We used RT-qPCR to validate the expression levels of the genes selected above. The gene expression level displayed significant differences in PM at different developmental stages. Among them, the relative expression of *Bcl2*, *Klf2*, *Klhl38*, *Idh3a*, and *Mylk2* in the PM was upregulated significantly (*p <* 0.01) (Figure 6A), whereas the relative expression of *Col22a1*, *Dnmt3b*, *Fn1*, *E2f1*, *Rprm*, and *Wfikkn1* were downregulated significantly (*p <* 0.01) (Figure 6B). Similar gene expression patterns were observed in the LM when using RT-qPCR. The Duck embryonic LM exhibited differential expression levels for *Idh3a*, *Got1*, *Bcl2*, *Mylk2*, *Klf2*, and *Klhl38* (*p <* 0.01) (Figure 6C). However, the difference in the relative expression level of *Erbin* was not significant (*p >* 0.05) (Figure 6C). Among the down-regulated genes, *Col22a1*, *Wfikkn1*, and *E2f1* showed significant differences (*p <* 0.01), whereas *Dnmt3b*, *Fn1*, and *Rprm* exhibited significant differences in LM at different developmental stages (*p >* 0.05) (Figure 6D).

### 2.7. Prediction of the CpG Islands on the Promoters

We used online software, MethPrimer (v1.0), to predict the existence of CpG islands on the promoters of the aforementioned genes. Among them, there were four CpG islands in the *Erbin* promoter region, with lengths of 104 bp, 118 bp, 182 bp, and 498 bp (Figure 7A). *Klf2* contained one CpG island with a length of 246 bp (Figure 7B). *Got1* appeared in four CpG islands, with total lengths of 110 bp, 107 bp, 107 bp, and 476 bp (Figure 7C). *Klhl38* contained two CpG islands with lengths of 123 bp and 129 bp (Figure 7D), and *Mylk2* included three CpG islands with lengths of 107 bp, 192 bp, and 341 bp (Figure 7E).

## 3. Discussion

The production of meat and meat quality influenced by skeletal muscles are important economic traits in duck breeding [11,18]. The development of skeletal muscles during the embryonic period determines the number and type of muscle fibers in poultry [3,9,19]. The process of muscle fiber hypertrophy to maturity that occurs at the terminal stage of embryonic development is undoubtedly crucial [4]. Although DNA methylation has been studied thoroughly in mammals [16,20,21], the systematic landscape of DNA methylation during the development of skeletal muscles in ducks remains largely unknown. In the present study, we examined the dynamic regulation of genome-wide DNA methylation in the development of skeletal muscles in embryonic ducks using WGBS and RNA-Seq technologies.

The proliferation, differentiation, and fusion of myoblasts were regulated by DNA methylation [4]. The DNA methylation profiles of Tan and Hu sheep and their offspring were analyzed systematically by using WGBS, and the data demonstrated that the DNA methylation of genes (*Acta1*, *Myh11*, *Was*, *Vav1*, *Fn1*, and *Rock2*) exerted important regulatory effects on differential muscle development in sheep [13]. It was reported that a group of myogenesis-related genes was collaboratively regulated by both 5mC and m6A modifications in postnatal skeletal muscle growth in pigs [14]. Similar to the DNA methylation patterns in sheep and pigs [13,14], the WGBS result of this study suggested that gene bodies were hyper-methylated, and the sites near the TSS showed the lowest levels in embryonic duck skeletal muscles. AMPK is a master regulator of metabolism, and its activity is essential in the regulation of energy available to muscle fibers by specifically regulating fatty acid oxidation pathways [22]. Previous studies showed that AMPK phosphorylation promoted atrophy and inhibited hypertrophy in skeletal muscles [7,23]. The significant enrichment of DMGs in metabolic pathways was observed in this study. Metabolism plays a crucial role in the late stage of development of duck skeletal muscles when muscle fibers are gradually hypertrophied and reach maturity. Therefore, we speculated that DMGs affect the metabolism and hypertrophy of skeletal muscle through DNA methylation.

A previous study revealed that the DNA methylation levels of individual genomic features were negatively correlated with gene expression levels according to an integration analysis of DNA methylation and transcription profiles [24]. The potential genes and pathways that mainly affect fatty acid metabolism and muscle development in poultry were identified using integrated DNA methylation with transcriptomes [25]. In this study, the negatively related genes were enriched in the KEGG signaling pathway associated with the development of skeletal muscles, showing the potential of DNA methylation to regulate myofiber formation and maturation in ducks. In Pigs, previous studies revealed that DNA methylation programming was involved in the regulation of myogenic differentiation and the phenotypic variation of skeletal muscles by mediating the MAPK signaling pathway [16,26]. Previous studies also showed that the inhibition of JNK/MAPK signaling by DNA methylation was involved in regulating myogenic differentiation [16]. The study showed that MAPK signaling was found to be involved in myofiber maturation through DNA methylation. *Apelin* affected mitochondrial biosynthesis through DNA methylation of *Ppargc1a*, and it then triggered hypertrophy [27]. Additionally, *Apelin* enhanced glucose uptake in skeletal muscles by activating Akt phosphorylation [27]. Similar to these results, *Apelin* influenced myofiber formation and hypertrophy by increasing mitochondrial biosynthesis through DNA methylation during the development of skeletal muscles in ducks. DNA methylation in mice regulated satellite cell differentiation through Wnt signaling [28]. *Klf5* was found to regulate skeletal muscle atrophy through the classical Wnt/β-catenin signaling pathway in chickens [29]. It can be inferred that the Wnt signaling pathway exerts its function through DNA methylation in myofiber hypertrophy in ducks. The insulin signaling pathway transmitted information to the intracellular signaling pathway PI3K/AKT to control muscle hypertrophy [30]. Previous studies found that insulin stimulated the DNA methylation of *Dapk3*, which led to autophagy and apoptosis [31]. In ducks, the methylation of gene promoters in the insulin signaling pathway affected glucose metabolism and regulated the development of skeletal muscles.

This study focused on 13 DMGs associated with skeletal muscle development. The subsequent CpG island prediction of the DMGs in the promoter identified five genes, identifying key sites in the promoters that regulated muscle development. *Mylk2* has been reported to regulate myogenesis by phosphorylating *Mef2c* [18]. In addition, *Mylk2* activates actin and myosin contraction to enhance contractile tone in skeletal muscles [32]. This study indicated that the DNA demethylation of *Mylk2* promoted gene expression, which may further induce the process of myofiber hypertrophy in ducks. *Erbin* modulated the Erbb activity that affected muscle innervations and inhibited muscle atrophy in the injury pathway [33]. *Klhl38* regulated muscle hypertrophy and atrophy through the Akt signaling pathway [34]. In ducks, it can be inferred that the DNA methylation level of *Erbin* can affect muscle hypertrophy through the Erbb signaling pathway. Moreover, the DNA methylation level of *Klhl38* can regulate skeletal muscle development by affecting the Akt signaling pathway, and the transcriptional target of *Klf2* was associated with myocyte fusion [35,36]. In contrast, this study found that *Klf2* played an important role in promoting myofiber formation and inhibiting autophagy. It was reported that *Got1* plays an important role in a range of metabolic pathways [37], and it may influence metabolism and skeletal muscle hypertrophy through DNA methylation in ducks. Our results suggested that the development of skeletal muscles in ducks was regulated mainly by the hyper-methylation or hypo-methylation of the key loci of DMGs. However, the epigenetic mechanisms that regulate these genes remain to be investigated.

## 4. Materials and Methods

### 4.1. Tissue Sampling

All procedures were implemented according to the Local Experimental Animal Care Committee and were approved by the ethics committee of Nanjing Agricultural University (Nanjing, China; SYXK-2019-00085). The resource population of hybrids from crossing Liancheng white ducks with Cherry Valley ducks was selected as the research object. Fertilized duck embryos were incubated under identical conditions. The incubation process required the careful control of temperature (38.1 °C) and humidity (70–75%). PM and LM were collected from E21 and E28 male duck embryos, with six replicates at each stage for further study.

### 4.2. DNA and RNA Extraction

Collected embryonic skeletal muscles were stored in liquid nitrogen immediately. RNA was extracted by lysis with Trizol (Invitrogen, Carlsbad, CA, USA), followed by agarose gel electrophoresis to determine its integrity, and then RNA purity was determined by using a Nanodrop ND2000 spectrophotometer (Thermo Scientific, Wilmington, DE, USA). Only identified RNA samples were allowed to proceed to the following sequencing. Genomic DNA was extracted from skeletal muscles using a Tissue Sample DNA Extraction Kit (Takara, Dalian, China), and its purity was determined by using a Nanodrop ND2000. Finally, DNA quality was determined by 1% agarose gel electrophoresis to analyze the degree of DNA degradation and RNA or protein contamination.

### 4.3. RNA-Seq

High-quality RNA was used to construct the libraries. Once the libraries were qualified, different libraries were assembled depending on the effective concentration and the amount of data targeted. CASAVA base recognition converted image data of the sequenced fragments that were measured by high-throughput sequencers to reads. Paired-end clean reads were aligned to the reference genome using HISAT2 v2.0.5. Finally, DESeq2 software (v3.17) was used to perform differential expression analysis.

### 4.4. Genome-Wide DNA Methylation Profiling

Genomic DNA was interrupted randomly to 200–300 bp using Covaris S220, followed by Bisulfite processing. After treatment, C without methylation became U. However, the methylated C remained unchanged, and PCR amplification was performed to add the adaptors to obtain the final DNA library. After library qualification, different libraries were subjected to high-throughput sequencing pooling according to the effective concentration and amount of target data amount. Four kinds of fluorescently labeled dNTPs were added, and DNA polymerase and adaptor primers were added to the sequencing flow cell for amplification to get the sequence information of the fragment to be measured. The analysis was focused on individual samples to study methylation level and density, functional gene region, and 2K upstream/downstream region. We analyzed DMRs as a genomic region where the methylation level differed significantly among different biological samples.

### 4.5. Gene Functional Enrichment Analysis

The cluster Profiler R package was used for GO enrichment analysis of DEGs, with correction for gene length bias. A corrected *q*-value < 0.05 was the threshold for significant enrichment of DEGs in GO terms. The database resource KEGG helped to establish the high-level function and utility of biological systems from molecular-level information (http://www.genome.jp/kegg/, accessed on 10 January 2023). We checked the statistical enrichment of DEGs in KEGG pathways by using the clusterProfiler R package.

### 4.6. Verification of Critical Gene Expression Level

After erasing genomic DNA, cDNA was synthesized using 5X All-In-One RT MasterMix (Applied Biological Materials Inc., Richmond, BC, Canada). RT-qPCR was used to validate the expression level of critical genes. The instructions were followed for each RNA sample and repeated three times to perform RT-qPCR using Bio-Rad CFX96 Real-Time PCR (Bio-Rad, Hercules, CA, USA) detection system with SYBR Premix Ex Taq Ⅱ (Takara, Beijing, China). The final PCR reactions contained 10 μL 2 X SYBR Premix Ex Taq II (Tli RNaseH Plus), 0.8 μL of forwards and reverse primer (10 mM), 2 μL first-strand cDNA and 6.4 μL H2O. Cycling parameters were 95 °C for 30 s, followed by 40 cycles at 95 °C for 15 s and 60 °C for 30 s. The relative expression level of mRNA was calculated using the 2^−ΔΔCt^ method. We used Primer Premier 5.0 software for the design of all primers, with detailed information available in Table 2.

### 4.7. CpG Island Prediction of Key Genes

The gene sequence information of ducks provided in the NCBI database was input into the online software names MethPrimer (MethPrimer-Design MSP/BSP primers and predict CpG islands-Li Lab, PUMCH (urogene.org)) for the prediction of CpG islands, accessed on 25 May 2023. MethPrimer identifies CpG islands in one or more nucleotide sequences. The ratio of the observed-to-expected number of GC dinucleotides patterns is calculated over a window (sequence region), which is moved along the sequence. The detection parameters were set as fragment length ≥ 200 bp, the content of GC ≥ 50%, and the ratio of CpG content to expected content ≥ 0.6.

### 4.8. Statistical Analysis

*t*-tests were performed using SPSS 25.0. Experiments were conducted with three biological replicates and three technical replicates. Relative expression levels were normalized to β-actin. Results are expressed as the mean ± SEM, and the ordinate represents the log10-transformed values. *p* < 0.05 was considered statistically significant. * *p* < 0.05, ** *p* < 0.01.

## 5. Conclusions

The DMRs involved in the development of skeletal muscle in embryonic ducks were identified by integrating genome-wide DNA methylation and transcriptome analysis systematically. We found 13 genes that were potentially relevant to the development of duck skeletal muscles. The screening of enriched genes suggested that hyper-methylation inhibited the expression of *Idh3a*, *Got1*, *Bcl2*, *Mylk2*, *Klf2*, *Erbin*, and *Klhl38*, and hypo-methylation stimulated the expression of *Col22a1*, *Dnmt3b*, *Fn1*, *E2f1*, *Rprm*, and *Wfikkn1*. The CpG islands of the promoter regions of five key genes were predicted, which included *Erbin*, *Klf2*, *Got1*, *Klhl38,* and *Mylk2*. We gained new insights into the complex relationship between DNA methylation in promoter regions and gene expression. We also hypothesized that the DMRs altered the expression level, which affected the subsequent transcription and translation. These results provide valuable information for the further study of the epigenetic mechanisms that control the development of skeletal muscles in ducks.

## Figures and Tables

**Figure 1 ijms-24-15476-f001:**
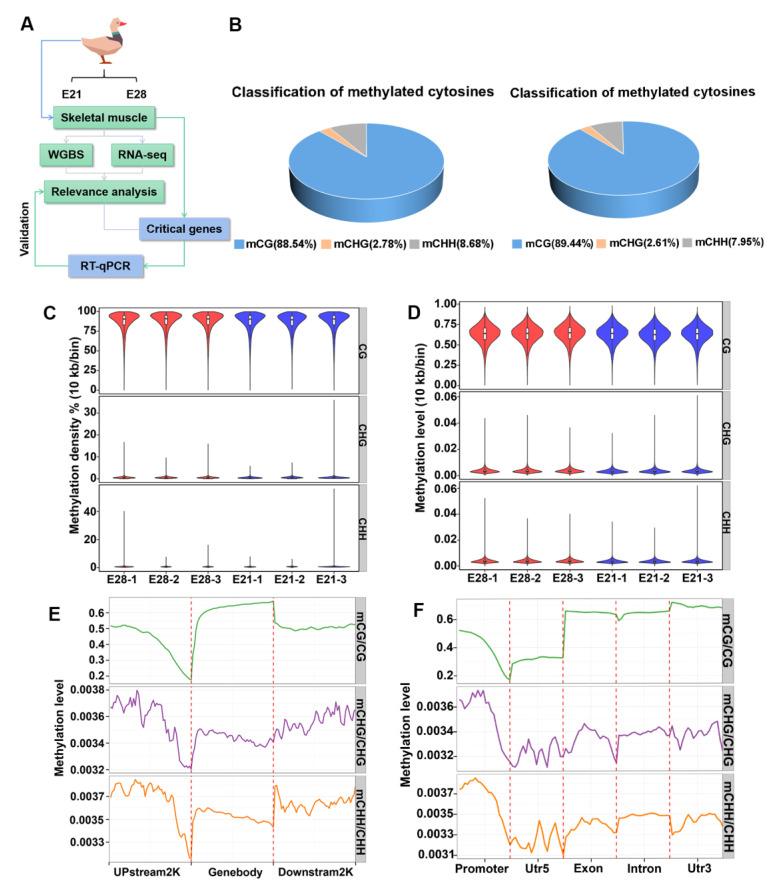
The profile of DNA methylation in embryonic ducks. (**A**) Experimental design. (**B**) The average proportion of different methylation types in the development of duck embryonic skeletal muscles. Methylated mCG, mCHG, and mCHH are represented by blue, orange, and grey, respectively. (**C**) Violin plots of the overall distribution of methylation levels across different methylation types. The right ordinate represents different contexts, which included CG and CHG and CHH, H = A, C, or T. The left ordinate represents the methylation level, and the abscissa represents different samples with 10 kb as a bin. (**D**) Violin plots of the overall distribution of methylation density for different methylation types. The right ordinate represents different contexts. The left ordinate represents the methylation level, and the horizontal co-ordinates represent different samples with 10 kb as a bin. (**E**) The distribution of methylation levels across different genomic elements. The abscissa shows genomic elements, and the ordinate displays methylation levels. The C site level of the functional regions of all genes was averaged, and different contexts were distinguished by different colors. (**F**) Distribution of methylation level at 2K upstream/downstream. Different regions are displayed on the abscissa, and the ordinate represents methylation levels. Different contexts are assigned different colors.

**Figure 2 ijms-24-15476-f002:**
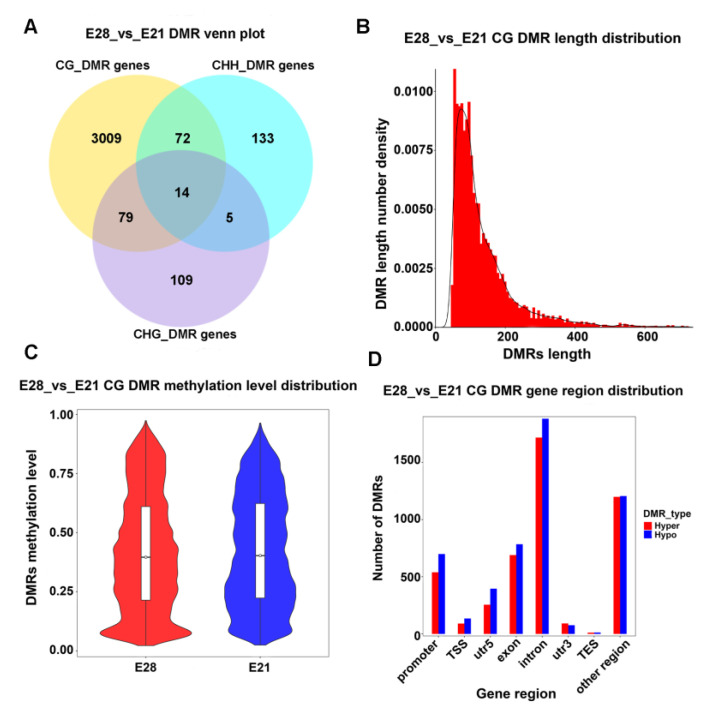
Identification of different methylation regions (DMRs) and different methylation genes (DMGs) in embryonic ducks. (**A**) Venn diagram of DMRs. (**B**) Length distribution of DMRs in the CG context. The DMR length is represented on the abscissa; the density at each length is represented on the ordinate, and the distribution of the fitted curve is shown in black. (**C**) Violin plot of the distribution of methylation level. Display of DMR methylation level distribution in the CG context. The horizontal axis represents the group of comparative portfolios, and the ordinate represents the methylation level value. (**D**) DMR anchor region in the CG context. Each region’s type is shown on the abscissa, and the number of hyper/hypo DMRs in each region is shown on the ordinate.

**Figure 3 ijms-24-15476-f003:**
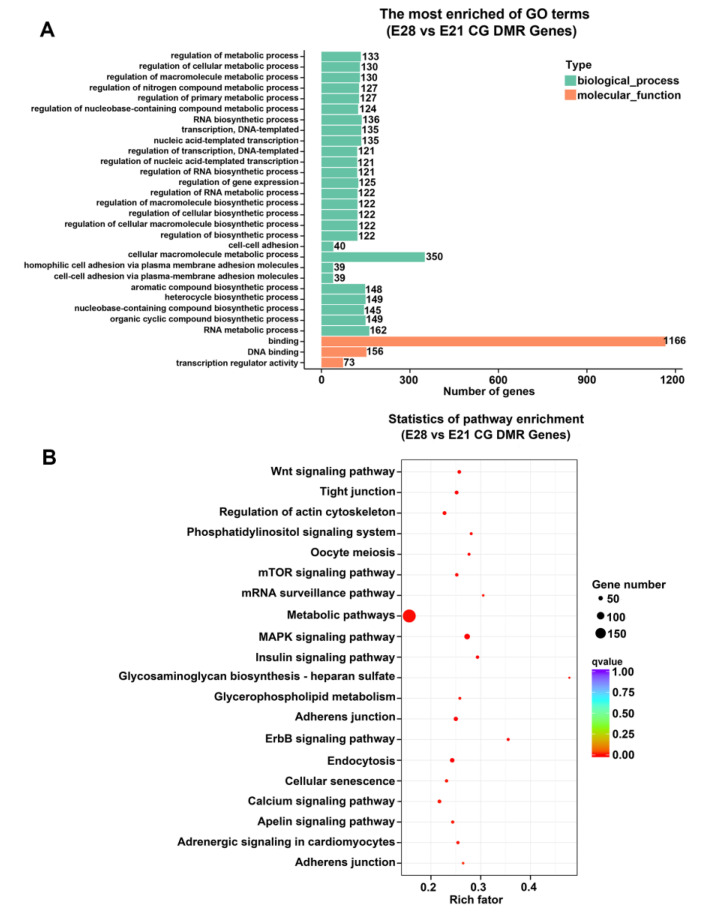
Functional enrichment analysis of DMGs. (**A**) Bar graph of the gene ontology (GO) enriched in the CG context. Classification statistics of enriched GO-related genes: The enriched GO term is plotted on the y-axis, and the number of DMR-related genes is on the x-axis. Biological processes and molecular functions are distinguished using various colors. (**B**) Scatterplot of Kyoto Encyclopedia of Genes and Genomes (KEGGs) metabolic pathways enriched in the CG context. The pathway name is represented on the vertical axis, and the rich factor is shown on the horizontal axis. The number of DMR-related genes in each pathway is denoted by the size of the points, and the color of the points corresponds to different *q*-values.

**Figure 4 ijms-24-15476-f004:**
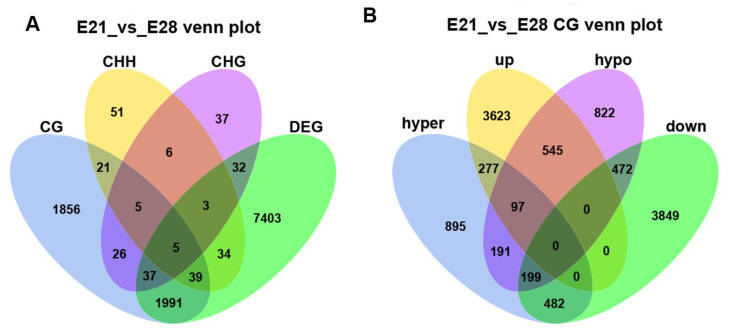
Association analysis of DNA methylation and gene expression. (**A**) Venn diagram of DMGs compared with DEGs. CG, CHG, and CHH represent DMGs in different contexts. (**B**) Venn diagram of DMGs compared with DEGs in the CG context. Hyper represents those genes anchored to the hyper-methylated regions, and hypo represents those genes anchored to the hypo-methylated regions; up represents highly expressed genes, and down represents lowly expressed genes.

**Figure 5 ijms-24-15476-f005:**
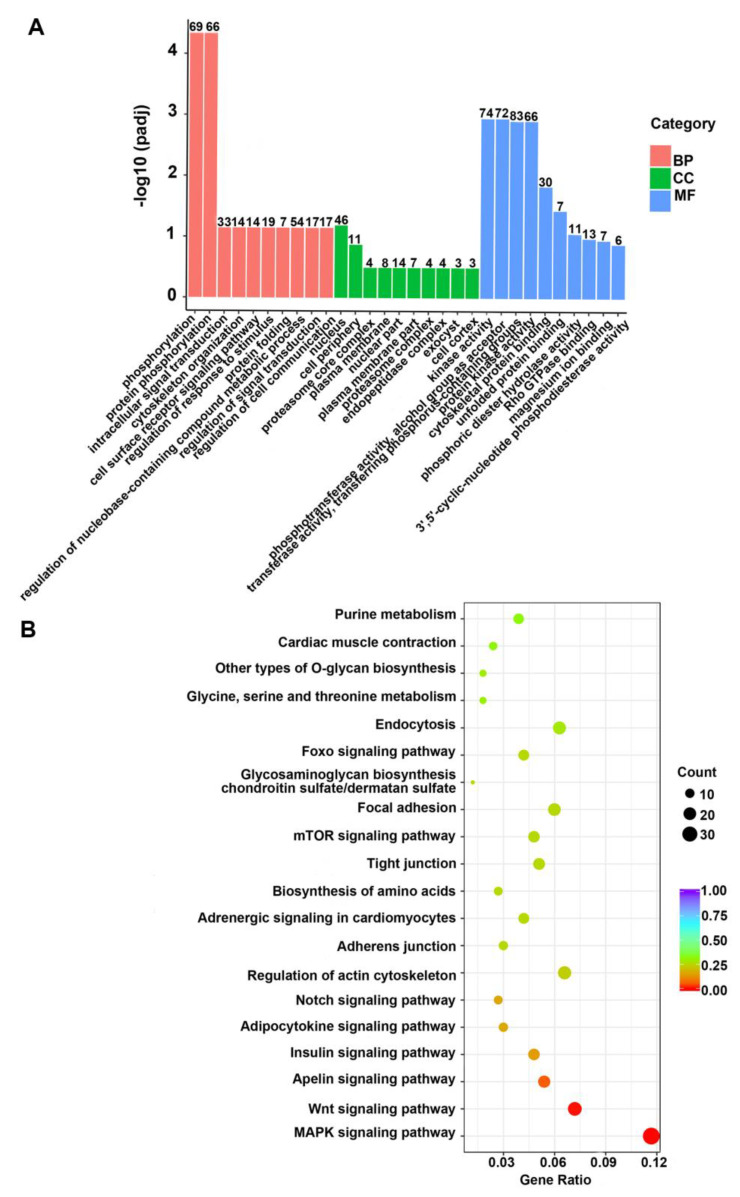
Functional enrichment analysis of the genes negatively associated with DMGs and DEGs. (**A**) The GO bar graph of the enrichment of the intersection genes. Classification of enriched GO-related genes: The enriched GO term is represented on the ordinate, and the number of DMR-related genes is shown on the abscissa. We use distinct colors to differentiate the biological processes, cellular components, and molecular functions. (**B**) Scatter plot of the KEGGs signaling pathways enriched for the intersection genes. The pathway name is represented on the vertical axis, and the rich factor is shown on the horizontal axis. The number of DMR-related genes in each pathway is denoted by the size of the points, and the color of the points corresponds to different *q*-values.

**Figure 6 ijms-24-15476-f006:**
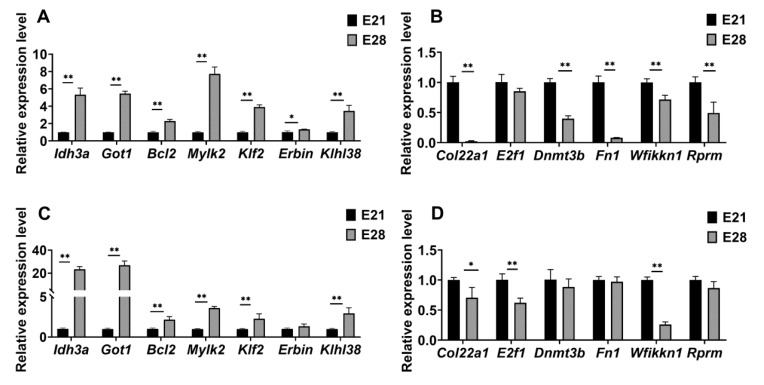
Quantitative validation of the vital genes. The relative expression of candidate genes at different developmental stages of skeletal muscle was determined by RT-qPCR. (**A**) DEGs upregulated and (**B**) downregulated in PM; (**C**) DEGs upregulated and (**D**) downregulated in LM. *β-actin* served as an internal reference. Values are expressed as the mean values ± SEM of the three replicates; * *p* < 0.05, ** *p* < 0.01.

**Figure 7 ijms-24-15476-f007:**
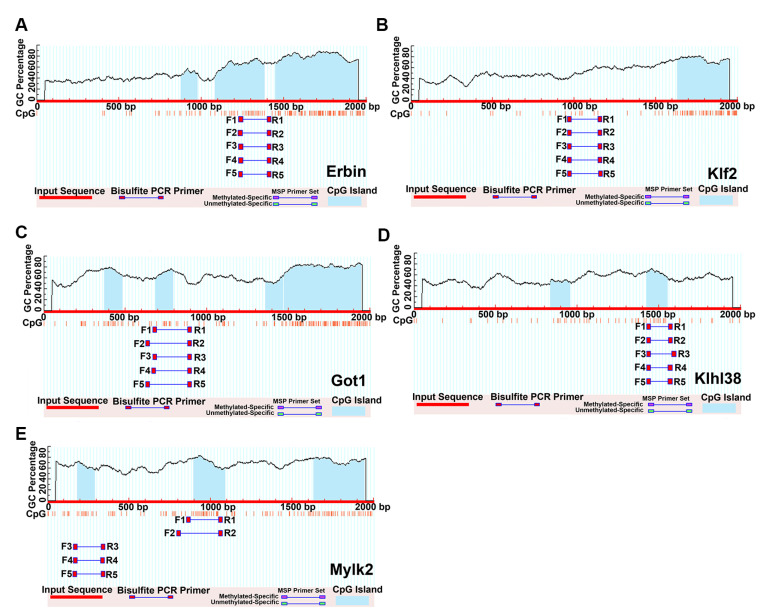
The prediction of the CpG islands in the promoters. CpG islands in the *Erbin* (**A**), *Klf2* (**B**), *Got1* (**C**), *Klhl38* (**D**), and *Mylk2* (**E**) promoter regions.

**Table 1 ijms-24-15476-t001:** Selected genes from the association analysis of the genome-wide DNA methylation and transcriptome.

Gene Name	Log_2_ Fold Change	Diff.Methy	C Context	Region
*Klhl38*	6.341923585	−0.161196584	CG	promoter
*Mylk2*	3.680553968	−0.353681294	CG	promoter
*Got1*	2.956657678	−0.209114374	CG	promoter
*Klf2*	1.683077721	−0.291024055	CG	promoter
*Erbin*	1.483720868	−0.310759514	CG	promoter
*Idh3a*	3.115334047	−0.126427439	CG	exon
*Bcl2*	1.555105211	−0.228206336	CG	exon
*E2f1*	−2.0475711	0.125491948	CG	exon
*Fn1*	−2.834137149	0.353384159	CG	exon
*Wfikkn1*	−3.111804698	0.161546317	CG	exon
*Dnmt3b*	−3.684262185	0.269994984	CG	exon
*Col22a1*	−4.123660418	0.35941219	CG	exon
*Rprm*	−5.226699842	0.122454333	CG	exon

**Table 2 ijms-24-15476-t002:** Primer information of RT-qPCR.

Gene Name	Primer Sequences	Product Length	Gene Accession Number
*Bcl2*	F: 5′-TGTGCGTGGAGAGCGTCAAC-3′	79	XM_027451680.2
	R: 5′-CGGTTCAGGTACTCGGTCATCC-3′		
*Klf2*	F: 5′-GCCTGCCCTTACAACCCTCTG-3′	172	XM_027472014.2
	R: 5′-TGCGAACTCTTGGTGTAGGTCTTC-3′		
*Fn1*	F: 5′-GGAGGTAGTCACAGTTGGCAATAC-3′	71	XM_038182473.1
	R: 5′-TGTCGTAGCAGGTGTCGTCAG-3′		
*Klhl38*	F: 5′-TCCAGGCAAGACAAGGCTACATC-3′	122	XM_038174946.1
	R: 5′-GCAAGTGGGTGAAGACTGAACAAG-3′		
*Erbin*	F: 5′-AGCAATATTCAGCCGGAAGC-3′	153	XM_038169994.1
	R: 5′-AACAAGTGCAGCCATCTGTG-3′		
*Mylk2*	F: 5′-AACGACACCGAGACGCTGAAC-3′	173	XM_038166457.1
	R: 5′-AGGTTGTTGAGCCAGGGATGC-3′		
*Dnmt3b*	F: 5′-TCTGGGGAAAGATCAAAGGTT-3′	191	XM_038166400.1
	R: 5′-AAACGTGGCAGAATTGAAATG-3′		
*Wfikkn1*	F: 5′-GGCTCTGACTGCGACATCTGG-3′	128	XM_027468956.2
	R: 5′-TCGGCGTCCATGTAGCACTTG-3′		
*Got1*	F: 5′-TTGCCGAGTGGAAGGACAACG-3′	138	XM_027460734.2
	R: 5′-AGCTGAACATGCCGATCTGGTC-3′		
*Idh3a*	F: 5′-TGCTGGATTGATTGGGGGTC-3′	146	XM_027465914.2
	R: 5′-CAGCACTCAGAAGAAGGGCA-3′		
*Col22a1*	F: 5′-ATTGACAAGTACGGCATACCACAG-3′	106	XM_038175073.1
	R: 5′-AAACCACTCTCACGGCATCTTTG-3′		
*Rprm*	F: 5′-TGGTGCAGATCGCCGTCATG-3′	73	XM_038182513.1
	R: 5′-AGGTTGCAGCCGAGGAAGAAG-3′		
*E2f1*	F: 5′-CTCGCTGAACCTCACCACCAAG-3′	113	XM_038166109.1
	R: 5′-AGATTCGCCTCTTCTGCACCTTC-3′		
*β-actin*	F: 5′-TGCGTGACATCAAGGAGAAG-3′	300	NM_001310421.1
	R: 5′-TGCCCGGGTACATTGTGGTA-3′		

## Data Availability

All WGBS and RNA-seq data are unpublished data from the Poultry Laboratory, College of Animal Science and Technology, Nanjing Agricultural University. The data presented in this study is available on request from the corresponding author.

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
