# Peer review of "The Integration of Genome-Wide DNA Methylation and Transcriptomics Identifies the Potential Genes That Regulate the Development of Skeletal Muscles in Ducks"

_ijms, 2023, doi:10.3390/ijms242015476_

Round 1
Reviewer 1 Report
The research presented in this manuscript is well planned and conducted, and the analysis was performed correctly. The positive aspect of this article is the combination of analyzes to produce results.
I don't know if the Journal requires expanding the abbreviations used for the first time in the body of the article, regardless of the abstract. If so, the abbreviations explained in the abstract are not explained again in the body of the article (starting with Introduction)
Not necessary spaces before dots at the end of some sentences.
Line 223 What kind of online software?
Line 250 CACNA2D2 - italic
Line 252 Ppargc1a - italic
Line 266 Erbb – italic
Why some genes you describe with capital letter and some with small letters?
Table 2 – change to one font character
Line 351 that13 – missing space
A significant drawback of the article is the lack of discussion of the results obtained with available literature. Most of the "discussion" chapter is introductory. Only this part can be considered a Discussion of obtained results, although it is only a collection of random sentences taken from literature reports: “It was reported that Mylk2 regulated myogenesis by phosphor-264 ylating Mef2c during the differentiation of myoblasts into myotubes [35]. Erbin localised 265 to the NMJ in skeletal muscles, which modulated gliadin-mediated Erbb activity, hence 266 participated in the disassembly of the AChR clusters [36]. The Sp1 attached Erk5 to Klf2 267 and the transcriptional target of Klf2 was associated with myocyte fusion [37].”
Well done research with no discussion - in my opinion, the discussion should be largely rewritten - it should focus on discussing the results obtained in the light of previous research
Author Response
We would like to thank the reviewers for their comments that will greatly improve the manuscript. We have studied comments carefully and have made correction. According to the comments, the changes were highlighted in red in the manuscript. Specific changes in the revised paper are listed below.
Reviewer #1:
Question 1. I don't know if the Journal requires expanding the abbreviations used for the first time in the body of the article, regardless of the abstract. If so, the abbreviations explained in the abstract are not explained again in the body of the article (starting with Introduction)
Response: Thanks for reviewer’s comments. According to requirements of Journal "Instructions for Authors", abbreviations should be defined when they first appear in each of the following three sections: the abstract; the main text; and the first figure or table. In response to this issue, the abbreviations were carefully revised.
Question 2. Not necessary spaces before dots at the end of some sentences.
Response: They were revised.
Question 3. Line 223 What kind of online software?
Response: Thanks for your questions. The online software used in this study was MethPrimer (MethPrimer-Design MSP/BSP primers and predict CpG islands - Li Lab, PUMCH (urogene.org)), and by inputting the duck gene sequence information provided in the NCBI database into the online software, it was possible to identify CpG islands in one or more nucleotide sequences. This information was added in Materials and Methods.
Question 4:Line 250 CACNA2D2 - italic
Line 252 Ppargc1a - italic
Line 266 Erbb – italic
Response: As suggested by the reviewer, they were revised.
Question 5. Why some genes you describe with capital letter and some with small letters?
Response: According to reviewer's suggestion, they were modified.
Question 6. Table 2 – change to one font character
Line 351 that13 – missing space
Response: As suggested by the reviewer, they were revised.
Question 7. A significant drawback of the article is the lack of discussion of the results obtained with available literature. Most of the "discussion" chapter is introductory. Only this part can be considered a Discussion of obtained results, although it is only a collection of random sentences taken from literature reports: “It was reported that Mylk2 regulated myogenesis by phosphor-264 ylating Mef2c during the differentiation of myoblasts into myotubes [35]. Erbin localised 265 to the NMJ in skeletal muscles, which modulated gliadin-mediated Erbb activity, hence 266 participated in the disassembly of the AChR clusters [36]. The Sp1 attached Erk5 to Klf2 267 and the transcriptional target of Klf2 was associated with myocyte fusion [37].”
Response: As suggested by the reviewer, discussion was carefully rewritten.
Reviewer 2 Report
A previous study by the group demonstrated a decline in 5hmC levels from d21 to d28 of duck muscle development, implying reciprocal TET demethylation activities. Thus, the authors compared DNA methylation with transcriptome analysis at the 2 time points to identify genes that may be developmentally regulated. Results demonstrate that hypermethylation correlates strongly with down-regulation of genes and vice versa. Thirteen selected genes were confirmed as DEGs by RT-PCR that may be involved in the transition from early to late secondary myogenesis. The selection process for the genes remains unknown as does their importance to muscle hypertrophy.
Major concerns
The manuscript is poorly written leading this reviewer to believe the authors are unfamiliar with developmental myogenesis. For example, the authors did not examine embryonic muscle formation but fetal myogenesis (d21 vs d28). The paper needs a serious rewrite in the context of muscle development during the later stage.
The greatest percentage of methylated genes are associated with metabolism. Why would the metabolic genes become downregulated during a period of rapid muscle hypertrophy (Fig 3)? This is an interesting finding that requires further discussion at minimum.
Paragraph 1 discusses the percentage of C methylation in CG, CHG and CHH context. CHG and CHH occurs in plants. The amount of methylation at these sites in the avian genome likely reflect background in the assay. What is the rationale for measuring CHG and CHH and what information can be gained from their analysis?
The authors confirm differential expression results from the integrated analysis for 13 genes. How and why were these genes selected? Known muscle genes that increase/decrease during this developmental window require measurement to place the selected genes into the context of developmental myogenesis. Numbers and size of myofibers and their morphology coupled with relative expression of Pax3/7, the MRFs, NFIX, TEAD4, etc. need to be measured.
The writing is highlighted by inappropriate use of terms, incorrect grammar, poor sentence structure and nonlinear development of paragraphs and thoughts. It requires restructuring by a native English speaker or a language editing service.
Author Response
We would like to thank the reviewers for their comments that will greatly improve the manuscript. We have studied comments carefully and have made correction. According to the comments, the changes were highlighted in red in the manuscript. Specific changes in the revised paper are listed below.
Question 1. The manuscript is poorly written leading this reviewer to believe the authors are unfamiliar with developmental myogenesis. For example, the authors did not examine embryonic muscle formation but fetal myogenesis (d21 vs d28). The paper needs a serious rewrite in the context of muscle development during the later stage.
Response: Thanks for this valuable comment. Base on the result of our previous study, the profiles of DNA methylation of skeletal muscles in the crucial stages (E21 and E28) in embryonic ducks was studied. During this period, myofibers gradually lose their ability to divide and fuse to form myotubes and transform into mature myofibers. At the terminal stage of embryonic development, myofibers exhibit hypertrophic growth. The information has been added accordingly in Introduction and Discussion.
Question 2. The greatest percentage of methylated genes is associated with metabolism. Why would the metabolic genes become downregulated during a period of rapid muscle hypertrophy (Fig 3)? This is an interesting finding that requires further discussion at minimum.
Response: Thanks for your questions. In the study, it was observed that DMGs were significantly enriched in metabolic pathways. Metabolism plays an important role at the end of skeletal muscle development in duck embryos when muscle fibers gradually hypertrophy and enter maturation. AMPK, a master regulator of metabolism, its activity is essential in regulating the energy available to muscle fibers by specifically regulating fatty acid oxidation pathways [1]. Activation of AMPK led to the up-regulation of the muscle-specific E-3 ubiquitin ligase, resulting in decreased C2C12 myotube cross-sectional area and skeletal muscle dysfunction in mice [2]. Previous studies have shown that phosphorylation of the metabolic pathway AMPK promoted skeletal muscle atrophy and inhibited hypertrophy [3]. Therefore, we hypothesized that DMGs may affect metabolism and skeletal muscle hypertrophy through DNA methylation.
References
- Kahn, B.B.; Alquier, T.; Carling, D.; Hardie, D.G. AMP-activated protein kinase: Ancient energy gauge provides clues to modern understanding of metabolism. Cell Metabolism 2005, 1, 15-25.
- Chen, X.; Ji, Y.; Liu, R.; Zhu, X.; Wang, K.; Yang, X.; Liu, B.; Gao, Z.; Huang, Y.; Shen, Y.; et al. Mitochondrial dysfunction: roles in skeletal muscle atrophy. Journal of Translational Medicine 2023, 21, 503.
- Mikhail, A.I.; Ng, S.Y.; Mattina, S.R.; Ljubicic, V. AMPK is mitochondrial medicine for neuromuscular disorders. Trends in Molecular Medicine 2023, 29, 512-529.
Question 3. Paragraph 1 discusses the percentage of C methylation in CG, CHG and CHH context. CHG and CHH occurs in plants. The amount of methylation at these sites in the avian genome likely reflect background in the assay. What is the rationale for measuring CHG and CHH and what information can be gained from their analysis?
Response: Thanks for your suggestion. It is well known that DNA methylation in animals occurs mainly in CG context. In Figure 1-B, it can also be seen that the DNA methylation of CG in duck skeletal muscles accounts for up to 88%. In contrast, the DNA methylation levels of CHH and CHG accounted for a low percentage. Therefore, we focused on the methylation levels of CG in the following study.
Question 4. The authors confirm differential expression results from the integrated analysis for 13 genes. How and why were these genes selected? Known muscle genes that increase/decrease during this developmental window require measurement to place the selected genes into the context of developmental myogenesis. Numbers and size of myofibers and their morphology coupled with relative expression of Pax3/7, the MRFs, NFIX, TEAD4, etc. need to be measured.
Response: Thanks for your suggestion. In this study, we first screened for TOP genes by comparing gene expression level (log2 Fold Change) and differential methylation level (diff.Methy). Subsequently, an in-depth study of TOP genes was carried out by reviewing literature and conducting pathway enrichment analysis. The study revealed that Got1, Idh3a, Dnmt3b were associated with metabolic pathways. Klf2, E2f1, Bcl2, and Rprm related to apoptosis and autophagy. Erbin related to NOD-like receptor signaling pathway. Mylk2 related to muscle contraction and mitochondrial autophagy. Fn1 related to actin cytoskeleton. Col22a1 related to muscle contraction and MAPK signaling pathway. Wfikkn1 could inhibit muscle growth inhibitor. And gene expression of Klhl38 ranked TOP1. Further, the screening of enriched genes showed that hypermethylation inhibited the expression of Idh3a, Got1, Bcl2, Mylk2, Klf2, Erbin, and Klhl38, and hypomethylation stimulated the expression of Col22a1, Dnmt3b, Fn1, E2f1, Rprm, and Wfikkn1. Therefore, these genes were screened as genes related to skeletal muscle development in ducks.
The mRNA expression of Pax7, MRFS and MyHC in E14, E21 and E28 duck skeletal muscles was measured by RT-qPCR analysis in our previous study [1]. The expression level of Pax7 in pectoral muscles increased during development, and the expression levels of Myf5, MyoD, MyoG and MyHC all reached peak at E21 and then decreased in E28. The morphology of pectoral muscle myofibers in duck embryos was also tested in E14-E28 pectoral muscles, and result showed that the density of myfibers gradually decreased from E21 to E28. Therefore, based on these previous results, we focused on whole genome methylation sequencing and transcriptome sequencing of duck skeletal muscles in E21 and E28.
References
- Li, D.F.; Ge, J.Y.; Wang, H.B.; He, Z.L.; Jin, T.; Yu, M.L. Differences in skeletal muscle development among Pekin duck, Liancheng white duck and their hybrid F_5 resource groups. Journal of nanjing agricultural university 2022, 45, 1235-1245.
Comments on the Quality of English Language
Question 1. The writing is highlighted by inappropriate use of terms, incorrect grammar, poor sentence structure and nonlinear development of paragraphs and thoughts. It requires restructuring by a native English speaker or a language editing service.
Response: Thank you very much for valuable advice. The manuscript has been edited for proper English language, grammar, punctuation, spelling, and overall style by the highly qualified native English speaker—— Thomas A. Gavin, Professor Emeritus, Cornell University.
Round 2
Reviewer 2 Report
Thank you for improving the writing and formatting of the manuscript. It reads better.
You MUST change the discussion of myogenesis in the Intro and Discussion from differentiation and fusion of MYOFIBERs to myoblasts. Muscle fibers are post-mitotic, multinucleated cells capable only of producing the requisite contractile machinery. Only mononucleated myoblasts are capable of proliferation and subsequent differentiation and fusion
much improved with a few minor tense issues present. the typesetter should detect and correct those.
Author Response
Question 1. You MUST change the discussion of myogenesis in the Intro and Discussion from differentiation and fusion of MYOFIBERs to myoblasts. Muscle fibers are post-mitotic, multinucleated cells capable only of producing the requisite contractile machinery. Only mononucleated myoblasts are capable of proliferation and subsequent differentiation and fusion.
Response: Thanks very much for reviewer’s valuable comments. The discussion related to myogenesis in the Introduction and Discussion was carefully revised.
Question 2. Comments on the Quality of English Language
much improved with a few minor tense issues present. the typesetter should detect and correct those.
Response: The typing errors in the manuscript were all corrected.